

# A network approach to identify bioregions in the distribution of Mediterranean amphipods associated with *Posidonia oceanica* meadows

Bruno Bellisario, Federica Camisa, Chiara Abbattista and Roberta Cimmaruta

Department of Ecological and Biological Sciences, Università degli Studi Della Tuscia, Viterbo, Italy

## ABSTRACT

Although amphipods are key components of the macro-fauna associated with *Posidonia oceanica* meadows, to date no studies focused on the structure and diversity of their assemblages across the whole Mediterranean Sea. Here, we applied a network approach based on modularity on a dataset mined from literature to identify biogeographic modules and to assess the biogeographic roles of associated localities. We also correlated the patterns evidenced with the biogeographic distribution of amphipod groups by means of a multivariate analysis. Modularity analysis highlighted four biogeographic modules bounded by the main Mediterranean biogeographic divides and evidenced a decrease in species diversity along a NW-SE gradient. Assemblages associated with Central-Western Mediterranean and, to a lesser extent, Tunisian modules showed the highest species richness and were identified as hubs, characterized by species with regional distributions that behave as source in a biogeographic context. The paleogeographic history of the host seagrass and the ecology of associated amphipods, both suggest the joint effect of species persistence and post-Last Glacial Maximum expansion in explaining the pattern of amphipod distribution in the Mediterranean Sea.

## INTRODUCTION

The Mediterranean Sea represents only 0.3% of ocean waters, yet it is a recognized hot-spot of biodiversity hosting about 17,000 species (*Coll et al., 2010*; *Bianchi et al., 2012*). This high diversity stems from a combination of oceanographic, ecological and biogeographic features allowing the coexistence of species of Atlantic origin with temperate and subtropical organisms (*Coll et al., 2010*). Latitudinal clines of environmental variables (mainly temperature and salinity) combined with marine currents resulted in a generalised latitudinal gradient of both primary production and species richness, decreasing from north-eastern to south-western regions (*Coll et al., 2010*; *Lejeusne et al., 2010*). Accordingly, biogeographic sectors were identified within the Mediterranean Sea, each characterized by both different biota and ecological parameters (*Bianchi et al., 2012*).

Corresponding author
Bruno Bellisario,
bruno.bellisario@unitus.it

Although general patterns of biodiversity distribution within the Mediterranean Sea have been well described (*Coll et al., 2010*; *Bianchi et al., 2012*), studies concerning patterns and mechanisms of species co-occurrence across the whole Mediterranean basin are restricted to a relatively small number of organisms (e.g., *Arvanitidis et al., 2002*; *Gerovasileiou & Voultsiadou, 2012*). Moreover, geographically widespread and ecologically broadly adapted groups have been only seldom studied according to habitat-related subdivisions (*Sevastou et al., 2013*). In this work, we focused on amphipod crustaceans associated with *Posidonia oceanica* (L.) Delile, 1813 meadows, since they represent one of the most relevant components of the vagile fauna of this key seagrass, endemic to the Mediterranean Sea.

*Posidonia oceanica* plays a fundamental role in ecosystem engineering along the Mediterranean coasts, providing important ecosystem functions, including oxygen production, food and shelter for associated species, as well as reduction of coastal erosion (*Boudouresque, Mayot & Pergent, 2006*). The complexity of this multi-layered and three-dimensional habitat allows a great variety of associated fauna to live into the canopy, rhizomes and mattes, making the meadows a strikingly biodiversity-rich habitat within the Mediterranean (*Buia, Gambi & Zupo, 2000*). Among the vagile fauna associated with meadows, amphipod crustaceans are one of the dominant groups, showing high abundance and diversity of species (*Mazzella, Scipione & Buia, 1989*; *Gambi et al., 1992*; *Sturaro et al., 2015*). Amphipods are key ecological components of seagrass habitats, due to their role in transferring energy across the system, and represent an important trophic resource for higher predators such as fish (*Pinnegar et al., 2000*; *Zakhama-Sraieb, Ramzi Sghaier & Charfi-Cheikhrouha, 2011*; *Michel et al., 2015*; *Bellisario et al., 2016*). Amphipods associated with *P. oceanica* meadows feed preferentially on macroepiphytes, algae and associated detritus (*Michel et al., 2015*), establishing a sort of facilitative interaction with the host plant by promoting the seagrass growth and obtaining protection against predation (*Valentine & Duffy, 2006*).

Despite the importance of amphipods in seagrass systems, a comprehensive study on their biogeographic patterns at the whole Mediterranean scale is still lacking. Available data include mainly check-lists and local studies based on classical diversity index (*Gambi et al., 1992*; *Diviacco, 1988*; *Como et al., 2008*; *Scipione & Zupo, 2010*; *Bedini et al., 2011*; *Zakhama-Sraieb, Ramzi Sghaier & Charfi-Cheikhrouha, 2011*; *Sturaro et al., 2015*), which can foster biogeographic studies using innovative approaches.

Recently, specific metrics rooted in network analysis have been successfully applied in a biogeographic context, outperforming classic approaches as clustering methods in the identification of bioregions (*Carstensen & Olesen, 2009*; *Vilhena & Antonelli, 2015*; *Bloomfield, Knerr & Encinas-Viso, 2018*). In particular, modularity (i.e., the tendency of a network to subdivide in densely connected modules or clusters) has proved powerful in detecting groups of areas and/or species closely connected together (i.e., biogeographical modules, sensu *Carstensen et al., 2012*; *Carstensen et al., 2013*). This approach also provides relevant insights into the processes driving the assembly of communities by evaluating the importance of each local assemblage (represented by nodes) in terms of network connectivity (*Bloomfield, Knerr & Encinas-Viso, 2018*). Specific metrics related to the number of links within and between biogeographic modules can be used as indirect

estimators of richness and endemism and provide information on the source/sink role of localities (*Carstensen et al., 2012*; *Carstensen et al., 2013*; *Bloomfield, Knerr & Encinas-Viso, 2018*).

In this work, we mined data from literature on the distribution of amphipods associated with *P. oceanica* meadows along the Mediterranean basin. We then used a network approach based on modularity: (1) to detect biogeographic structure; (2) to correlate the patterns evidenced with the current knowledge on the biogeographic distribution and ecological features of amphipod groups; (3) to compare amphipods diversity with the paleogeographic history of Mediterranean Sea *P. oceanica* seagrass. The results obtained are discussed with the aim to provide insights on the patterns of amphipod diversity and distribution across the Mediterranean basin.

## MATERIALS & METHODS

### Study area and starting dataset

An extensive survey of the literature was conducted to obtain all available information on the presence of amphipods from *P. oceanica* meadows across different regions of the Mediterranean Sea. We filtered the available literature to obtain comparable data in terms of sampling season, depth and methods (see Supplemental Information 1), so recovering data from 11 papers (*Diviacco, 1988*; *Scipione et al., 1996*; *Scipione, 1998*; *Sánchez-Jerez, Carberá Cebrian & Ramos Esplá, 2000*; *Zakhama-Sraieb, Ramzi Sghaier & Charfi-Cheikhrouha, 2006*; *Zakhama-Sraieb, Ramzi Sghaier & Charfi-Cheikhrouha, 2011*; *Bedini et al., 2011*; *Sturaro et al., 2014*; *Sturaro et al., 2015*; *Bellisario et al., 2016*; *Camisa et al., 2017*). Data were checked for possible taxonomic issues by updating species nomenclature according to WoRMS, so that species names reported in Table S1 correspond to present day taxonomic assignment (see Supplemental Information 1 for further explanations). The final dataset (available as Supporting Dataset) included 147 amphipod species from 28 localities: nine located in Tunisia, 16 in Italy (Tyrrhenian, Adriatic and Ionian Sea), two in Spain and one in Corsica (France). The literature source for each locality and each species is listed in Supplemental Information 1. Available data covered a large portion of the Mediterranean basin (Fig. 1), and were distributed in regions characterized by different geographic, hydrological and geological features, as well as by differences in the potential connectivity due to the general circulation models (*Bianchi & Morri, 2000*; *Bianchi, 2007*; *Berline et al., 2014*).

Information about the biogeographic distribution of observed amphipods were obtained from *Bellan-Santini & Ruffo (2003)*, which classified over 400 species of Mediterranean benthic amphipods in twelve macro-categories on the basis of their current distribution: WM, West Mediterranean; EM, East Mediterranean; Adr, Adriatic Sea; ME, Mediterranean endemics; Afr, African coasts from Ceuta to Cap Vert; Ib, Iberian coasts; Fr, French coasts; Br, British coasts; Norw, Norwegian coasts; Arct, Arctic Sea; Ind-P, Indo-Pacific Ocean; Cosm, Cosmopolitan. Here, species with an Atlantic distribution were clumped in two main categories from the five proposed by *Bellan-Santini & Ruffo (2003)*: ATL, Atlantic Sea (Iberian, French and British coasts) and NATL, North Atlantic Sea (Norwegian and Arctic regions), so that our final distribution comprised nine different categories.

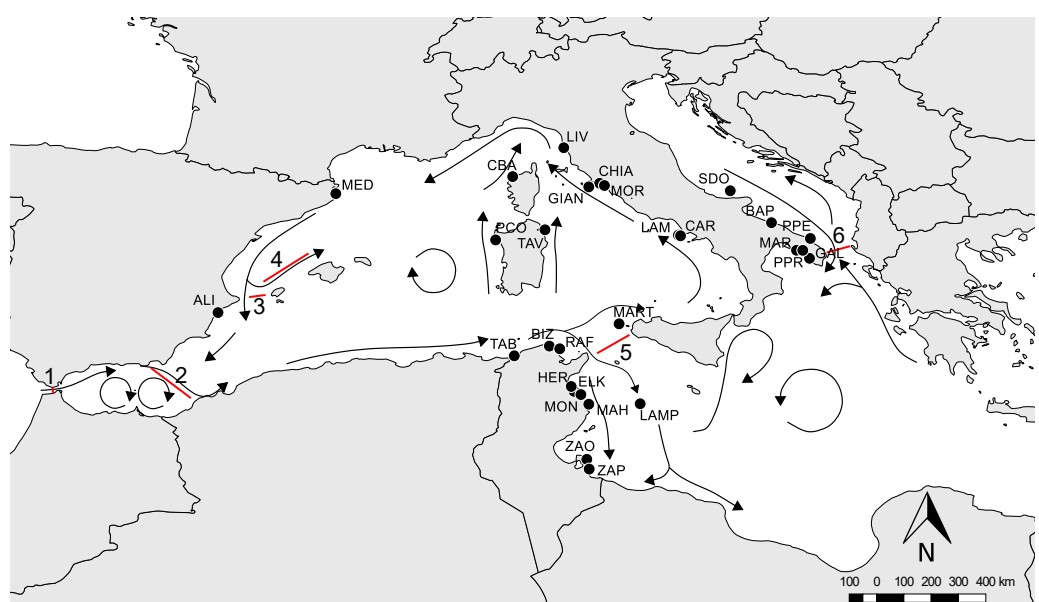

**Figure 1** **Overview of the Mediterranean Sea with reference localities.** Arrows indicate the main circulation patterns and red lines the main barriers: 1, Gibraltar Strait; 2, Almeria-Oran Front; 3, Ibiza Channel; 4, Balearic Front; 5, Sicily Channel; 6, Otranto Strait. For acronyms, see Supplemental Information 1.

## Network analysis

To provide insights into the biogeographic distribution of amphipod assemblages, a thresholding approach was used to identify groups of localities having stronger similarity in terms of community composition (*Kivelä, Arnaud-Haond & Saramäki, 2015*).

Data were ordered as a species/incidence matrix, whose entries represent the presence of species (rows) at each locality (columns). A weighted network, where localities represent nodes and links the strength of connections, was built using the Jaccard index on the species/incidence matrix, to derive a similarity distance matrix of species co-occurrence. Values ranged from 0, when two localities were identical in amphipod composition, to 1, when they shared no taxa, so that links with higher weights indicated low similarity between localities, and vice versa. The thresholding approach was then applied to identify closely related localities by finding the critical value describing the threshold similarity among pairs (i.e., percolation network).

Percolation networks are becoming increasingly used in ecological studies since they allow identifying relationships among nodes (i.e., populations, species, critical scales in landscape ecology) with the advantage of not requiring any *a priori* knowledge of a threshold value (*Rozenfeld et al., 2008*; *Fletcher Jr et al., 2013*; *Bellisario, 2018*). This value was measured by removing distances in decreasing order (i.e., most dissimilar localities), until the network reached the threshold value beyond which it becomes fragmented into disconnected clusters. The identification of this value is obtained by calculating the average cluster size $< L >$ that is, the average number of localities belonging to an $l$-size cluster, as a function of the last threshold distance value beyond which links were removed (*Stauffer*

*& Aharony, 1992*):

$$< L > = \frac{1}{N} \sum_{l < l_{\max}} l^2 n_l \tag{1}$$

where $N$ is the total number of localities not included in the largest cluster ($l_{\max}$) and $n_l$ is the number of clusters containing $l$ localities. Basically, each time a distance value is removed from the network, localities are redistributed in clusters of different sizes, from largest to smallest. This procedure is therefore iterated until the critical threshold is identified in the transitional region characterized by a strong decrease in $< L >$ where the network becomes disconnected (for more information about percolation theory, refer to *Stauffer & Aharony (1992)*. Here, we used the methodology described in *Rozenfeld et al. (2008)* and implemented in the package 'sidier' (*Muñoz Pajares, 2013*) of R (*R Development Core Team, 2018*).

## Modularity

After identifying the minimum set of pairwise similarities between localities, we tested for the presence of a significant pattern of aggregation between localities, and if this pattern reflected a geographic component. To this end, we measured the modularity ($Q$), which is defined as the degree to which a network can be subdivided in aggregated sets of nodes (i.e., modules), where the within-module links are significantly higher than between-module ones (*Newman & Girvan, 2004*; *Fortunato, 2010*). Modularity provides a formal description of the pattern of aggregation between species, populations or communities, being able to identify critical scales in specific ecological and evolutionary processes (*Fletcher Jr et al., 2013*).

Modularity was measured by using the equation originally described by *Newman & Girvan (2004)*:

$$Q = 1/2m \sum_{i,j} [A_{ij} - P_{ij}] \delta(C_i, C_j) \tag{2}$$

where $m$ is the total number of links in the percolation network (see above), $A_{ij}$ is the matrix expressing the degree of similarity between localities $i$ and $j$, $\delta(C_i, C_j)$ is a matrix indicating whether $i$ and $j$ are members of the same module and $P_{ij}$ is the probability in the null model that a link exists between $i$ and $j$. The extent to which links are distributed within and among modules was tested against an appropriate null model, to correct the observed value of $Q$ by null model expectation. Here we used a simulated annealing algorithm (SA) to test for the significance of a modular partitioning by generating 1,000 null matrices having the same degree distribution as the original network. Under the SA algorithm, affiliation of nodes to modules has an accuracy of 90%, and a significant modular structure was found if the empirical $Q$ value lies above the 95% confidence interval for $Q$ in the randomized networks (*Guimerà & Amaral, 2005*).

Starting from the modular partition, we further assigned the role of each locality in the network by using two topological measures related to the number of species of the local fauna ($l$, local topological richness) and the distribution of its associated species to other modules ($r$, regional topological linkage) (*Carstensen et al., 2012*). The two-dimensional

space given by *l-r* allows the subdivision of localities in: peripherals, few local and regional species; non-hub connectors, few local and many regional species; provincial hubs, many local and few regional species; connector hubs, many local and regional species (revised after *Carstensen et al., 2012*). Following *Carstensen et al. (2012)*, non-hub localities (i.e., peripherals and connectors) can be interpreted as sink, able to receive species from source localities both within their own module and of other modules. Conversely, hub-localities (i.e., provincial and connector hubs) can be interpreted as source for both their modules (module hubs) and the entire network (network hubs). As links in our network relate with patterns of similarity between assemblages, the role of localities allowed for a straightforward description of how amphipod diversity could have spread between different areas of the Mediterranean basin.

### Multivariate analysis

To explore to what extent the measured network characteristics (i.e., modularity and nodes topology) were related to the biogeographic distribution of amphipods, we ran a between-group correspondence analysis (BGCA) on the 'sites × species-biogeographic classes' matrix, where groups were given by the identified modules. BGCA performs a classic Correspondence Analysis (CA) of the per-group centers of gravity, providing an ordination of the groups by maximizing the between-group variance (*Baty et al., 2006*). From the nine species-biogeographic classes derived from the literature (see above), data were aggregated by summing the number of species belonging to each class at a given site.

## RESULTS

The final dataset obtained from literature showed that 147 amphipod species belonging to 77 genera have been identified to date in *P. oceanica* meadows from 28 Mediterranean localities (Supplemental Information 1).

### Modules identification

The percolation network showed a co-occurrence similarity threshold of 0.74, which means that sampling sites are expected to share no more than 74% of amphipod species. This leaded to a network structure of 28 localities joined by 104 links, showing a significant modular structure when compared with randomized models ($Q = 0.466 \pm 0.005$, $P < 0.001$). Four distinct modules were identified, characterized by a clear geographic distribution: Central-Western Mediterranean (CWM), Tunisian (TUN), Ionian (ION) and Adriatic (ADR) (Fig. 2A).

   Eleven localities constituted the Central-Western Mediterranean module (CWM), which spanned from the Spanish to southern Tyrrhenian coasts, including Lampedusa (LAM) and Marettimo (MART) Islands. This module contained the highest number of species (130), of which a high percentage were module exclusives (i.e., present in a single module). Eighty species were exclusively linked (i.e., observed) to the CWM module, which shared 20, 11 and two species with the TUN, ION and ADR modules, respectively, while 14 were in common among all the four modules (Fig. 2B). The CWM module also showed the highest number of Mediterranean endemics (sensu *Bellan-Santini & Ruffo,*

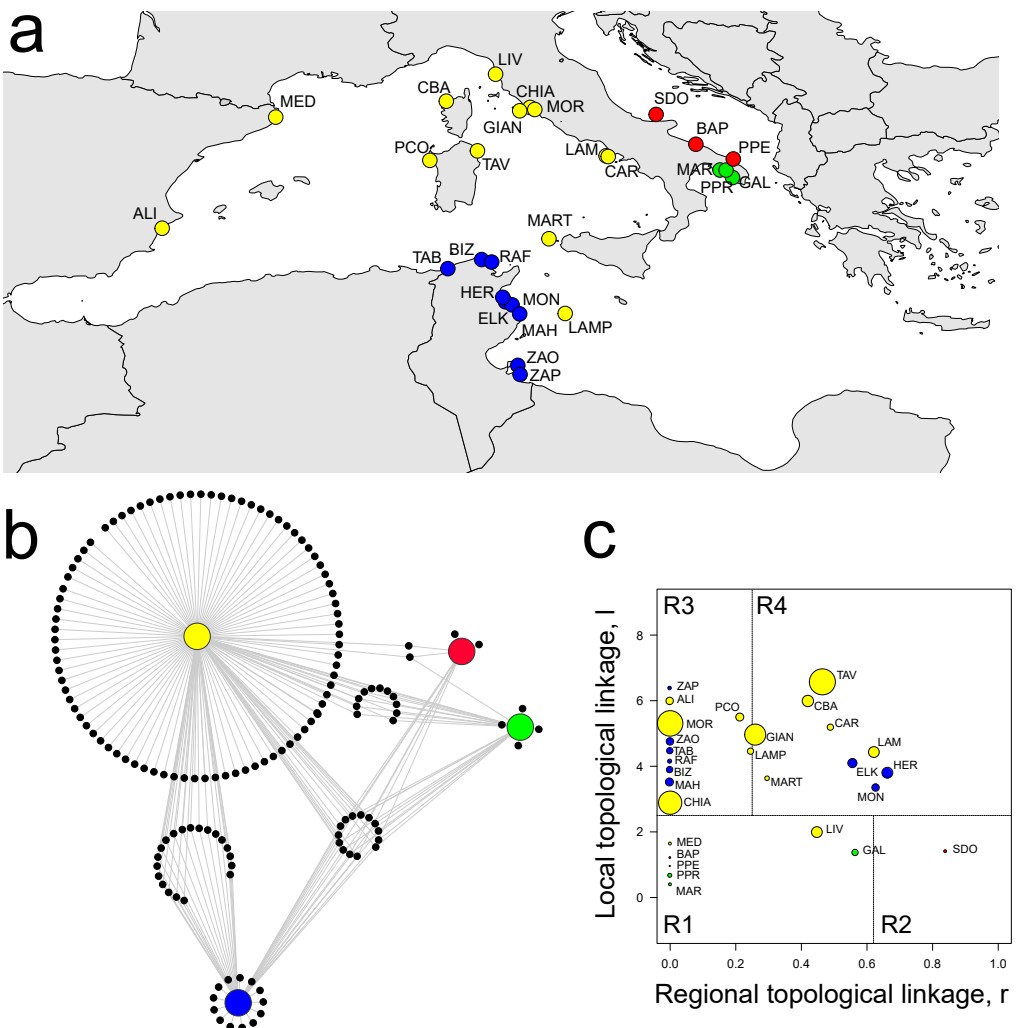

**Figure 2** **Spatial distribution of modules, network structure and biogeographic roles of localities.** (A) Spatial distribution of the four modules identified: yellow, CWM; blue, TUN; green, ION; red, ADR. (B) Network visualization of biogeographic modules with associated amphipod species (black dots). (C) Plot showing the biogeographic role of localities. Coordinates *l* and *r* describe the species richness of localities and the geographical distribution of associated species, respectively. Node size is proportional to species diversity. R1, peripherals: few local and regional species; R2, non-hub connectors: few local and many regional species; R3, provincial hubs: many local and few regional species; R4, connector hubs: many local and regional species (revised after *Carstensen et al., 2012*). For locality acronyms, see Supplemental Information 1.

*2003*) as, for example, species belonging to the genus *Peltocoxa: P. gibbosa* (Schiecke, 1977), *P. mediterranea* Schiecke, 1977, *P. marioni* Catta, 1875 (Fig. 2B, Table 1 and Table S1).

The Tunisian module (TUN) comprised all nine localities belonging to the Tunisian coasts and showed both a high number of species and a high percentage of module exclusives, for example the species belonging to the genus *Elasmopus: E. brasiliensis* (Dana, 1855), *E. pectenicrus* (Spence Bate, 1862), *E. pocillimanus* (Spence Bate, 1862). This module
**Table 1  Modular subdivision of sampling localities.** *L* is the number of localities in each module; *s* the total number of species. Module exclusive is the percentage of species exclusively present in a single module; ME and COSMP are the percentage of species in each module belonging to the Mediterranean Endemics (ME) and Cosmopolite (COSMP) biogeographic classes (see Supplemental Information 1).

| Module | L | s | Module exclusive (%) | Mediterranean endemics (ME%) | Mediterranean cosmopolite (COSMP%) |
|---|---|---|---|---|---|
| CWM | 11 | 130 | 64.7 | 16.7 | 4.8 |
| TUN | 9 | 46 | 27.6 | 7.3 | 7.3 |
| ION | 3 | 26 | 15.4 | 13 | 4.3 |
| ADR | 3 | 15 | 18.4 | 9 | 27.3 |

was also characterized by the lowest number of Mediterranean endemics (Fig. 2B, Table 1 and Table S1).

Both the Adriatic (ADR, three localities) and Ionian (ION, three localities) modules were characterized by having a few species and a low percentage of module exclusives (Fig. 2B and Table 1). The ION module showed some Mediterranean endemics like *Iphimedia minuta* G. O. Sars, 1883 or *Maera pachytelson* Karaman & Ruffo, 1971, while the ADR module showed a high percentage of cosmopolitan species.

The topological role of localities has been assessed by modularity analysis (Fig. 2C), so that each locality has been assigned to a category according to the topological linkage, i.e., local or regional. Localities with a few, local (i.e., module exclusive) species are considered as peripheral nodes (R1 in Fig. 2C), while nodes with a high number of species characterized by regional distribution (i.e., shared among many modules) are considered as connector hubs (R4 in Fig. 2C). Localities in the ADR and ION modules were all classified as peripherals or non-hub connectors (R1 and R2 in Fig. 2C), meaning that amphipod assemblages in these localities are composed by few local species and by a higher (although not very consistent) number of species having a regional distribution (see 'Materials & Methods'). Localities in the CWM and TUN modules were classified mainly as hubs, subdivided between provincial and connector hubs (R3 and R4 in Fig. 2C). More than half (54%) of localities in the CWM module can be considered connector hubs, characterized by many local and regional species, while most of localities in the TUN module were classified as provincial hubs, so having a larger number of local than regional species (Fig. 2C). The highest values of species diversity are found in hub localities (R3 and R4), while the lowest values of diversity were found in peripheral localities (R1 and R2) together with some hubs, as the 9 TUN localities (Fig. 2C).

## Multivariate pattern

The first two principal dimensions of the BGCA accounted for almost 80% of the total inertia (Fig. 3), showing that some biogeographic groups of species concur in explaining the observed pattern of between-modules variance (Fig. 3). Mediterranean endemic species (ME) accounted for almost 50% of the total variance explained by the first dimension of the BGCA, which was associated with the pattern of distribution of assemblages in the CWM module. Indo-Pacific (INDP) and African (AFR) species accounted for more than 30% of variance along the first dimension (Fig. 3), providing a clear differentiation of the TUN module from all others. Cosmopolitan species (COSMP) accounted for

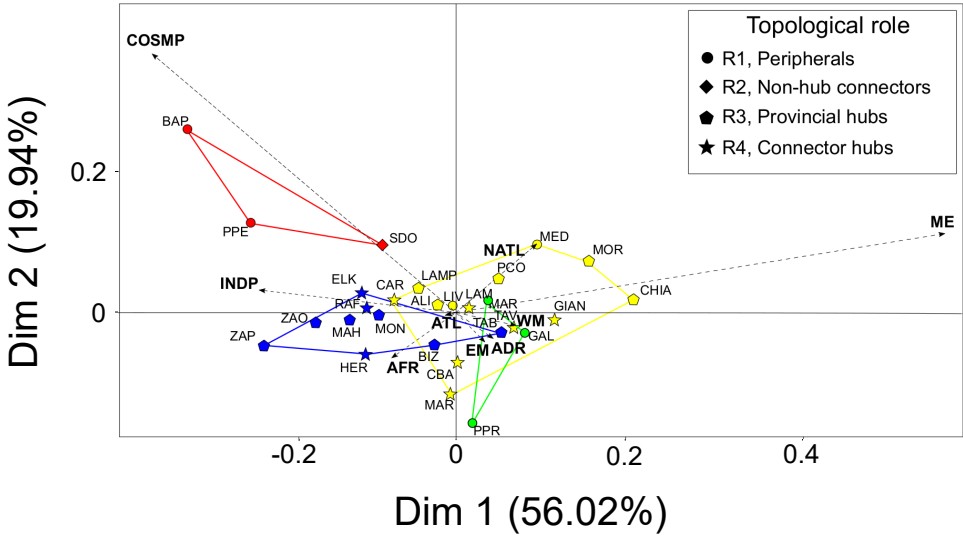

**Figure 3** **Between-Group Correspondence Analysis (BGCA) of localities based on the nine species-biogeographic classes according to literature.** Colours indicate the identified modules (see Fig. 2A) and different shapes correspond to the biogeographic role of localities (see top-right box). Dashed arrows indicate the nine species-biogeographic classes (revised after *Bellan-Santini & Ruffo, 2003*): COSMP, Cosmopolitan; INDP, Indo-Pacific; AFR, African coasts from Ceuta to Cap Vert; ATL, Atlantic coasts from Spain to Britain; NATL, North Atlantic from Norway to Arctic Sea; WM, West Mediterranean; EM, East Mediterranean; ADR, Adriatic Sea; ME, Mediterranean endemics. For locality acronyms, see Supplemental Information 1.

20% of the between-module variance explained by the second dimension of the BGCA, characterizing the pattern of ordination of the ADR module (Fig. 3). With respect to the biogeographic role of localities, provincial and connector hubs seemed to be characterized by both Mediterranean endemics (ME) and species of Indo-Pacific distribution (INDP), while peripheral localities were characterized mainly by cosmopolitan species (COSMP), although the overall pattern was not sharply defined (Fig. 3).

# DISCUSSION

## Modules are biogeographically based

Our findings showed that amphipod assemblages are heterogeneous throughout the Mediterranean area, with a maximum of 74% of species shared among localities, and that these differences lay on a geographical base. Despite the network approach does not relay on spatial information, the identified modules correspond to four geographic regions of the Mediterranean Sea: Central-Western Mediterranean (CWM), Tunisian (TUN), Ionian (ION) and Adriatic (ADR). Each region is delimited by well-known barriers, such as the Almeria-Oran Front, the Sicily Channel and the Strait of Otranto (highlighted as n. 2, 5 and 6 in Fig. 1). All these barriers have been pointed out as the most relevant in accounting for by ecological and biogeographic heterogeneity across the Mediterranean Sea, and all of them set a quite abrupt change in salinity and temperature regimes of adjacent basins.

The Almeria-Oran Front (AOF) is the western boundary of the Central-Western module (CWM), corresponding to the range boundary of the Mediterranean endemic *P. oceanica*, unable to tolerate the low temperature and salinity of the Alboran Sea (*Boudouresque, 2004*). Circulation patterns and changes in temperature and salinity across the Sicily Channel, both concur in partially preventing the dispersal of a number of species across the threshold of the Siculo-Tunisian Straits (*Robinson et al., 1992*; *Coll et al., 2010*). In our study, this is the divide between the CWM and TUN modules, with this latter grouping the localities along the Tunisian coasts characterized by the presence of the jet-like Algerian Current and Atlantic Ionian Stream (*Pinardi & Masetti, 2000*). The Strait of Otranto delimits the Adriatic Sea, a semi-enclosed basin where several factors, as winds, tides and freshwater runoff from rivers, all determine peculiar low salinity and low winter temperatures (*Falco et al., 2000*; *Lejeusne et al., 2010*). This is the boundary between the ADR and ION modules, which moderately exchange water mass through the Albanian side of the Strait of Otranto (*Orlic, Gacic & Laviolette, 1992*).

The boundaries between modules are represented by the most effective Mediterranean barriers, in agreement with the geographic patterns highlighted in other organisms studied at the whole basin scale. The areas corresponding to the Western and Eastern Mediterranean and to the Adriatic Sea have been historically considered as different biogeographic provinces, hosting differentiated species assemblages of macrophytes, diatoms and many animal groups (*Ignatiades et al., 2009*; *Gambi, Lampadariou & Danovaro, 2010*). Among invertebrates, an analysis of the Mediterranean sponge regional diversity showed distinct assemblages from CW Mediterranean, Tunisia, Adriatic and Ionian Sea (*Gerovasileiou & Voultsiadou, 2012*). Similar results were reported for benthic polychaetes, showing different assemblages in the Western and Central Mediterranean basins and in the Adriatic Sea (*Arvanitidis et al., 2002*), while deep-sea megafauna showed dissimilar community compositions in Western Mediterranean and Ionian basins (*Tecchio et al., 2010*).

## Modules diversity and species distribution

The differentiation among the four modules depends on different aspects of assemblage diversity and composition. The assemblages characterized by higher diversity were all from the CWM module, and in particular were located in the Central Tyrrhenian area (e.g., CHIA, MOR, GIAN, TAV; Fig. 2B). Localities from ADR and ION modules showed the lowest diversity values, together with a few localities from CWM (MART, MED) and TUN (ZAP, RAF). A decreasing gradient in species richness from north-west to south-east is a generalized pattern in the Mediterranean Sea, with an overall animal species diversity 100% greater in the western than in the eastern basin in both vertebrates and invertebrates (*Boudouresque, 2004*; *Coll et al., 2010*). As an example, the diversity of deep-sea nematode assemblages decreases with depth but, when similar depths are compared, a longitude effect appears, with diversity decreasing eastward (*Danovaro et al., 2008*). A similar pattern was detected in deep-sea foraminifers, whose species richness decreases from western to eastern Mediterranean, likely mimicking the longitudinal cline of organic matter availability on the deep seafloor (*Danovaro et al., 2010*).

The biogeographic role of localities showed how assemblages differ among and within modules, by identifying hubs *vs.* peripheral localities. Localities with assemblages having a high number of species characterized by regional distribution (i.e., shared among many modules) are considered as connector hubs and are supposed to behave as source in a biogeographic context. On the opposite, localities where assemblages contain few, local species are labeled as peripheral nodes, and considered as sinks. Interestingly, nearly all the localities of CWM and TUN modules are classified as hubs, meaning that they share a large proportion of species with many other localities belonging to other modules, besides a relevant number of species among each other. On the contrary, all the localities from ADR and ION modules are considered as peripherals or ultra-peripherals, so having assemblages very similar at intra-module level and highly differentiated with respect to localities belonging to other modules. Both modules are characterized by a small number of module exclusive species, together with a sub-set of species found either in the whole basin or in the hub localities of the CWM module.

The identified modules differ not only in species richness and biogeographic roles of their associated localities, but also in the biogeographic distribution of the species found in various assemblages. The pattern recovered links species' richness and distribution, with low-richness modules characterized by species having wider distributions and vice-versa, which can be explained in the light of biogeographic considerations. For instance, Indo-Pacific and African species associated with warm waters typify the rich Tunisian assemblages, as for example those belonging to the genus *Elasmopus* (Fig. 3). Mediterranean endemics and, to a lesser extent, North-Atlantic species characterize the high diversity of CWM. *Apherusa chiereghinii* Giordani-Soika, 1949, *Cressa cristata* Myers, 1969, *Gammaropsis crenulata* Krapp-Schickel & Myers, 1979, and the species belonging to the genus *Peltocoxa* are examples of Mediterranean endemics of the CWM module, while some species belonging to the genus *Ampelisca*, like *A. serraticaudata* Chevreux, 1888 and *A. tenuicornis* Lilljeborg, 1855, represent an example of temperate species of Atlantic origin inhabiting also the western part of the Mediterranean Sea. This pattern is due to the well-known paleogeographic history of the basin, with particular regard to the most recent cycles of Plio-Pleistocene glaciations (*Coll et al., 2010*). These climatic oscillations induced temperate Atlantic species to periodically enter the Mediterranean Sea where they originated vicariant endemic species as a result of geographic isolation and local adaptation. Examples are species belonging to the genera *Apherusa, Tritaeta* and *Tmetonyx*, which represent the cold component of the Mediterranean amphipod fauna (*Bellan-Santini & Ruffo, 2003*). For these reasons the Mediterranean Sea is considered as a 'diversity pump' from the Atlantic and the identification of its biogeographic provinces largely rely on the distribution of Mediterranean endemics (*Bianchi & Morri, 2000*; *Bianchi et al., 2012*). Similarly, it has been shown that species originated from warm faunas prevail in the south-eastern part of the Mediterranean basin (*Lejeusne et al., 2010*). Accordingly, the presence of warm species as a representative of the Tunisian coasts has been signaled for other invertebrates, e.g., sponges (*Gerovasileiou & Voultsiadou, 2012*).

Assemblages in the geographically confined ADR module are characterized mainly by widely distributed species with cosmopolitan range. This observation, and the fact that

only few species are found exclusively in this module, both suggest that this area may be particularly difficult to be colonized and behave as a sink, as shown by modularity analysis (Fig. 3C). This difficulty can be related to both extreme environmental conditions and geographic isolation, so that only vagile and tolerant species may enter and establish in this basin. Indeed, the Adriatic Sea is characterized by low salinity and winter temperatures, together with a moderate water mass exchange with the neighbouring Ionian Sea through the Strait of Otranto (*Orlic, Gacic & Laviolette, 1992*; *Falco et al., 2000*). Moreover, the Adriatic basin was largely dried during the Last Glacial Maximum (LGM, 23K-19K years ago; *Rohling et al., 2010*), so that a significant part of the Adriatic fauna has only recently re-colonized this basin. Within the poorly vagile group of amphipods, which are brooding species lacking a pelagic larval stage, cosmopolitan species are generally euryhaline, eurytherm and more prone to passive dispersal (*Bellan-Santini & Ruffo, 2003*), thus having the highest probability to reach and settle in the Adriatic habitat. Accordingly, a recent checklist of opisthobranch Adriatic fauna signaled that the great majority of species had an Atlantic-Mediterranean range, while only few were Mediterranean endemics (*Zenetos et al., 2016*).

## Amphipod diversity and paleogeographic history of *Posidonia oceanica*

Populations of *P. oceanica* inhabiting the western and eastern parts of the Mediterranean Sea are genetically differentiated, with those from the central Mediterranean around the Siculo-Tunisian Strait characterized by a higher genetic diversity (*Arnaud-Haond et al., 2007*; *Serra et al., 2010*). This pattern was interpreted as the result of a secondary contact zone between the western and eastern forms, each one originated by vicariance in glacial refugia during the Last Glacial Maximum (LGM) (*Serra et al., 2010*). A more recent study has superimposed Ecological Niche Modelling to phylogeographic data, highlighting the southern Mediterranean as the most climatically suitable area during LGM, with particular regard to the central zone. This area was then proposed as the main glacial refugium of the seagrass, thus explaining its higher genetic diversity as due to the long-term persistence in this region (*Chefaoui, Duarte & Serrão, 2017*).

Glacial refugia have been repeatedly pointed out as hot-spots and melting pots of diversity, not only in terrestrial environments but also in marine habitats (*Hewitt, 1999*; *Hewitt, 2004*; *Maggs et al., 2008*). This diversity may concern both genetic lineages and community richness, in agreement with the postulated relationship between habitat stability and community diversity (*Hewitt, 2000*; *Ives & Carpenter, 2007*). Within this frame, the high diversity and the hub role of assemblages in TUN module can be explained by their localization in a glacial refuge area, as proposed by *Chefaoui, Duarte & Serrão (2017)*. However, the same pattern characterized the CWM assemblages, which are richer in species and share an even greater percentage of their species with other modules. This finding, however, may be only apparently in contrast with the lower presence probability of *P. oceanica* in the northern Mediterranean during LGM, if we consider that amphipods living on *P. oceanica* are not exclusively found in this habitat.

Indeed, amphipods can actively choose their substratum, and this habitat preference produces differences in their abundance on various seagrasses and other substrates, rather than a presence/absence pattern (*Sanchez-Jerez, Barberá-Cebrián & Ramos-Esplá, 1999*; *Vázquez-Luis, Sanchez-Jerez & Bayle-Sempere, 2009*). A possible scenario is therefore that the northward seagrass range expansion triggered by climate warming after LGM prompted the migration of part of the associated fauna. At the same time, the north-western Mediterranean was likely already inhabited by a local pool of amphipod species derived from the Atlantic (according to the 'biodiversity pump' mechanism), therefore adapted to temperate climatic conditions and able to survive during LGM. Under this hypothesis, it is expected that the CWM assemblages would include many module endemics represented by species with Mediterranean and/or Atlantic distribution (i.e., ME and ATL according to biogeographic classes). Also, CWM and TUN modules should share a relevant number of species (i.e., those originated in the southern refuge and migrated northward with the seagrass). Our analyses showed that all these expectations were verified, supporting the blending between resident, temperate species with the warm-adapted ones spreading from the south with *P. oceanica* during LGM in explaining the current richness of amphipod fauna in CWM.

## CONCLUSIONS

To our knowledge, this study is the first trying to assess the pattern of co-occurrence of Mediterranean benthic amphipod assemblages associated with a peculiar habitat, such as *P. oceanica* meadows. A network approach based on modularity has proven useful in detecting the biogeographic subdivisions of assemblages and in assessing biogeographic roles of associated localities. Our results provide a new perspective on the less studied southern Mediterranean, which may gain a relevant place in the origin of the basin biodiversity as a LGM refuge, besides confirming the known role of the Central Western Mediterranean area as a 'biodiversity pump' from the Atlantic. Our findings also suggest how the distribution of amphipod diversity in *P. oceanica* meadows stems from complex interactions between present and past geographic barriers, local species adaptation, and the biogeography of the host plant.

To this end, comparative phylogeographical studies can be used to verify the assumptions made in this study, in order to clarify the history of the biogeographical modules identified. This can open the road to a series of studies aimed at deepening the knowledge of the most common and abundant species shared by modules, to identify the centre of evolution and/or dispersion of amphipods within the Mediterranean. For instance, a comparative phylogeography between congeneric species shared by the TUN and CWM modules but showing different distributions (e.g., species belonging to genera *Apolochus, Ampithoe, Dexamine*) can help both disentangling the role of the southern basin in the diversity of amphipods and testing for the hypothesis of a convergence between the ecological and evolutionary outcomes of amphipods and seagrass in the Mediterranean basin.

## ACKNOWLEDGEMENTS

The authors would like to thanks the Academic Editor and Referees whose suggestions greatly improved the final outcome of the manuscript. We also would like to thanks Prof. Michele Scardi for all stimulating conversations during the first draft of the manuscript.

### Funding

The authors received no funding for this work.

### Competing Interests

The authors declare there are no competing interests.

### Author Contributions

- Bruno Bellisario conceived and designed the experiments, performed the experiments, analyzed the data, contributed reagents/materials/analysis tools, prepared figures and/or tables, authored or reviewed drafts of the paper, approved the final draft.
- Federica Camisa and Chiara Abbattista performed the experiments, contributed reagents/materials/analysis tools, approved the final draft.
- Roberta Cimmaruta conceived and designed the experiments, analyzed the data, contributed reagents/materials/analysis tools, authored or reviewed drafts of the paper, approved the final draft.

### Data Availability

The raw data are available in File S2.

### Supplemental Information

Supplemental information for this article can be found online at http://dx.doi.org/10.7717/peerj.6786#supplemental-information.

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
