# Peer review of "A network approach to identify bioregions in the distribution of Mediterranean amphipods associated with Posidonia oceanica meadows"

_PeerJ, doi:10.7717/peerj.6786_

## Round 0.1 · original submission · Major Revisions

· Academic Editor

Major Revisions

While both reviewers agree that your work has interesting and important aspects, both also mention the need for 1) more concrete information on the species you are studying or looking at, and 2) improved English. I agree with their assessment, and would like to see lists or information on the species or taxa included in your study. As well, the writing includes many long and winding sentences that impede the ability of readers to clearly understand what you wish to say. In many cases, sentences can be rewritten into two sentences. Please consider your text carefully. As this will need much rewriting, my decision is that major revisions are needed.

I look forward to receiving a revised version of your work.

·

Basic reporting

English overall good, but a few passages are hard to read or grammatically flawed:

line 335
line 396
line 413
line 372: Relationships with Posidonia oceanica history - relationship of amphipods with...?

Experimental design

no comment

Validity of the findings

no comment

Additional comments

It is a brave and interesting effort, to bring so much data together and the scope of the paper is clear and indeed fills a gap in the knowledge in biographic classes in amphipods. However, in some respects the outcome remains rather abstract. About the hypothesis referred to in line 413 – this hypothesis or scenario is so broad in its predictive value that it makes almost every conceivable mode of distribution plausible.

I would like to see more reference to certain groups of amphipods referred to in line 413. Which amphipods dependent on Posidonia would be possible candidates in a follow up research to verify the hypothesis and why. It would certainly help strengthening the paper if more attention is given to this in the conclusions.

Reviewer 2 ·

Basic reporting

I feel that this manuscript needs major revision.

In my opinion, the entire manuscript is poorly structured, with long unclear statements. I feel that words are used incorrectly, such as "rate of endemisms" in the introduction. The authors keep referring to a "network approach based on modularity." I feel that this should be explained in the introduction as it is essentially jargon to anyone who is unfamiliar with these techniques. As it is, this article is relevant to a small group of researchers.

The figures are too busy and not explained well. There is no description of the labels.

Finally, the article title is "Biogeography of Mediterranean amphipods associated with Posidonia oceanica meadows", yet there is no mention of a single amphipod species in the article. I feel that this is crucial to the article and should not be supplementary information.

Experimental design

The research question is vaguely defined with jargon, and needs clarification. I am still unsure of exactly where the data came from.

Validity of the findings

I am sure the findings are valid and may be useful, but I feel that they should absolutely be presented in terms of what amphipod species are where. It is impossible to describe the biogeography of amphipods without doing this.

Additional comments

I think this manuscript is interesting, but needs to be presented differently, with more of a focus on the amphipods and less of a focus on your "network approach based on modularity."

---

## Round 0.2 · Major Revisions

· Academic Editor

Major Revisions

I have heard back from the same two reviewers as in round 1. Reviewer 2 has concerns about the literature and methodology used in your species identifications (and thus the identity of some of species), and as they are amphipod experts, I must concur with their assessment that your paper still needs work in this area.

My suggestions would be to 1) reconsider your title, and 2) list all species authorities, and also all literature you have utilized in making your identifications. In other words, you can implement some of Table S2 into your text, and/or include all references/authorities used to identify specimens in at least Table S2, if not in your text. By doing so, your methods will become replicable. Even if the reviewer does not ultimately agree with your identifications, at least then the methodology and logic are traceable and viewable to readers.

Thus, in summary, my decision is major revisions are needed, although it should be noted these revisions are not nearly as encompassing as the first round of reviews, and I anticipate you can resubmit sometime soon.

·

Basic reporting

The readability has much improved. I added or removed an 's' here and there on the track changes version which staff will forward to the authors.

Experimental design

no comment

Validity of the findings

The conclusions gained focus by pointing out relevant amhipod goups.

Reviewer 2 ·

Basic reporting

This version of the manuscript is much better than the first, but I still have several suggestions for improvement (see comments in PDF file). The authors need to add in text taxonomic authorities for all species mentioned and these also need to be added to the reference list. I feel that the title is misleading and perhaps should be changed to reflect that you are actually reporting on your model rather than the biogeography of amphipods. In my opinion you still do not discuss the biogeography of amphipods, but instead are reporting on your modeling using amphipods as a model group.

Experimental design

I think that the experimental design is sound, except for the literature used for data mining.

Validity of the findings

I have a concern as to the validity of your data as it is based on literature that is apparently not current (although I still feel that the sources of data are unclear). For example, you mention Leucothoe spinicarpa as a cosmopolitan species, yet it is not cosmopolitan and is not believed to be present in the Mediterranean Sea (see Krapp and Menioui, 2005). My concern for your results is that they are based on literature that may or may not be currently relevant.

Additional comments

There are still several grammatical and reference errors with the manuscript.

Annotated reviews are not available for download in order to protect the identity of reviewers who chose to remain anonymous.

---

## Round 0.3 · accepted · Accept

· Academic Editor

Accept

I have gone over the new version myself, and am satisfied that the paper is now acceptable for publication. Your efforts to list references and species authorities have improved the paper immensely, and the work can now be considered replicable. I look forward to seeing the final version!

#